# Does Lower-Limb Tendon Structure Influence Walking Gait?

**DOI:** 10.3390/healthcare11243142

**Published:** 2023-12-11

**Authors:** Alejandro Núñez-Trull, Javier Álvarez-Medina, Diego Jaén-Carrillo, Alberto Rubio-Peirotén, Ferrán Abat, Luis E. Roche-Seruendo, Eva M. Gómez-Trullén

**Affiliations:** 1Departamento de Fisiatría y Enfermería, Universidad de Zaragoza, 50009 Zaragoza, Spain; info@podologiazaragoza.es (A.N.-T.); javialv@unizar.es (J.Á.-M.); 2Department of Sport Science, University of Innsbruck, Innrain 52, 6020 Innsbruck, Austria; diego.jaen@uibk.ac.at; 3Universidad San Jorge, Campus Universitario, Autov A23 km 299, 50830 Villanueva de Gállego, Zaragoza, Spain; leroche@usj.es; 4Grup de Recerca GRACIS (GRC 01604), Sports Orthopaedic Department, ReSport Clinic, Universitat Pompeu Fabra, Escola Superior de Ciències de la Salut TecnoCampus, 08002 Mataró, Barcelona, Spain; abat@resportclinic.com; 5iHealthy, Research Group, Departamento de Fisiatría y Enfermería, Universidad de Zaragoza, 50009 Zaragoza, Spain; evagomez@unizar.es

**Keywords:** biomechanics, gait, lower limb, rocker, tendon morphology

## Abstract

Background: Within the exploration of human gait, key focal points include the examination of functional rockers and the influential role of tendon behavior in the intricate stretch–shortening cycle. To date, the possible relationship between these two fundamental factors in the analysis of human gait has not been studied. Therefore, this study aimed to analyze the relationship between the morphology of the patellar and Achilles tendons and plantar fascia with respect to the duration of the rockers. Methods: Thirty-nine healthy men (age: 28.42 ± 6.97 years; height: 173 ± 7.17 cm; weight: 67.75 ± 9.43 kg) were included. Data of the rockers were recorded using a baropodometric platform while participants walked over a 10 m walkway at a comfortable velocity. Before the trials, the thickness and cross-sectional area were recorded for the patellar tendon, Achilles tendon and plantar fascia using ultrasound examination. The relationship between the morphology of the soft tissue and the duration of the rockers was determined using a pairwise mean comparison (t-test). Results: A significant difference was found for rocker 1 duration, where a longer duration was found in the group of subjects with thicker patellar tendons. Regarding the Achilles tendon and plantar fascia, no significant differences were observed in terms of tendon morphology. However, subjects with thicker Achilles tendons showed a longer duration of rocker 1. Conclusions: The findings underscore a compelling association, revealing that an increased thickness of the patellar tendon significantly contributes to the extension of rocker 1 duration during walking in healthy adults.

## 1. Introduction

Tendons are critical connective structures within the human body, primarily composed of various collagen types, constituting a substantial proportion of their composition, ranging from 65% to 80% [1]. They play a pivotal role in various locomotor activities, such as running and walking, by functioning as springs that continuously compress and decompress during the movement of the lower limbs [2]. In these dynamic actions, tendons demonstrate their remarkable ability to store and release energy efficiently [3]. Tendons can be categorized as load transmitters and movement facilitators, with the Achilles tendon (AT) and the patellar tendon (PT) emerging as primary load-bearing structures in the lower limbs [4]. The plantar fascia (PF), closely associated with the AT due to its common insertion at the calcaneus bone, contributes to the load transmission mechanism and enhances its functionality [5]. Given the integral role of tendons in these activities, understanding their structure is pivotal for comprehending and improving human locomotion. In this context, two key morphological features, cross-sectional area (CSA) and thickness, have garnered significant attention in tendon research [6]. These features are typically examined using ultrasound (US) techniques [7]. Investigating these structural aspects of tendons can provide valuable insights into their functional capabilities and how they contribute to human movement.

The gait cycle, a fundamental aspect of human locomotion, is conventionally divided into two primary phases: stance and swing [8]. The swing phase, accounting for approximately 40% of the cycle, occurs when the foot is not in contact with the ground and the leg swings forward. It initiates with the toe-off (TO) event, where the foot leaves the ground, and concludes with the initial contact (IC) of the same foot. The stance phase, constituting roughly 60% of the cycle, occurs when one foot remains in contact with the ground. This phase begins with the IC event, typically marked by the heel’s contact with the ground, and ends with the TO event. The stance phase can also be further described in terms of four functional rockers, each with a distinct fulcrum [9].

The first rocker comes into play during the IC event, with the heel serving as the fulcrum, facilitating the rotation of the foot and enabling forward body movement [10]. The second rocker operates during mid-stance, with the ankle passively dorsiflexed, and it acts as the fulcrum while the limb moves over the foot [10]. As the stance phase progresses, the third and fourth rockers take over, with the fulcrum shifting to the metatarsal heads. The mid-tarsal joints lock, transforming the foot from a flexible structure into a rigid lever capable of propelling the body forward. The fourth rocker, often referred to as the toe-only rocker, supports the weight-bearing portion of the foot closest to the metatarsal heads, providing a stable midstance and reducing toe shock during TO [10]. Clinicians widely employ portable floor-level baropodometric platforms for gait analysis and quantification on flat surfaces in various populations [11,12,13]. However, despite their potential significance for clinicians in distinguishing between pathological and non-pathological gait, the evaluation of functional rockers has received limited attention from the scientific community.

It is well established that in certain pathological conditions, such as diabetic foot conditions or idiopathic toe walking, gait biomechanics are altered [14]. Patients with diabetic foot conditions often exhibit conservative gait strategies, including a slower walking speed, a wider base of gait, and prolonged double support time [14]. Additionally, these patients display differences in the lower-limb tendons, with increased thickness in the AT and PF [15]. While these alterations have been explored in subjects with the aforementioned pathologies and previous studies have also determined the impact of these soft tissues on motion features and the potential risk of foot injury [16,17], to the authors’ knowledge, the influence of the lower-limb tendon structure on the functional rockers in healthy individuals remains uncharted. Understanding this influence among healthy subjects holds the potential to unlock the key to determining if the observed tendon alterations in patients with pathological gait contribute to the gait anomalies they exhibit.

Therefore, the primary objective of this study is to investigate the relationship between the morphological characteristics of the PT, AT, and PF tendons and the duration of functional rockers during walking. The hypothesis underpinning this investigation posits that greater CSA and thickness of these tendons will be associated with shorter durations of the functional rockers. This study aims to address a critical knowledge gap in the field, shedding light on how tendon structure influences the mechanics of human locomotion and its potential implications for gait abnormalities in pathological conditions. Furthermore, it has the potential to inform clinical practice and the development of targeted interventions to enhance mobility and quality of life for individuals with gait disorders.

## 2. Materials and Methods

An observational study was conducted, which received approval from the local bioethics committee (009-19/20).

### 2.1. Participants

Thirty-nine healthy men (age: 28.42 ± 6.97 years; height: 173 ± 7.17 cm; weight: 67.75 ± 9.43 kg), older than 18 years and free from lower-limb injuries in the last 6 months, were voluntarily enrolled in this study. Participants with a pathological gait were excluded. Before starting, informed consent following the World Medical Association’s Declaration of Helsinki (2013) was signed. A sample large enough to represent the target population of this study was selected (Table 1).

Sample size power calculation was executed using G*POWER 3.1.9.7 (University of Dusseldorf, Dusseldorf, Germany). The following structure was used based on the analysis: mean differences between two independent means (matched pairs); a priori. Effect size dz = 0.5; α error prob = 0.05; power (1-β error prob) = 0.90. The result showed a suitable total sample size of 38 athletes for actual high power (90.25%).

### 2.2. Procedures

A one-session protocol was completed. Participants were asked to walk over a 10 m walkway at a comfortable velocity [18]. They started walking two meters from the recording space and stopped two meters behind the platform to minimize both acceleration and deceleration effects. Participants turned around and walked back to the starting point when they reached the end of the walkway. This procedure was repeated for 3 min, so that the subject walked on the examination walkway as many times as possible at a comfortable speed. Data from one step were collected for analysis using the Freemed™ baropodometric platform, collecting all the steps within the active sensor area.

### 2.3. Material and Testing

For descriptive purposes, the body height (cm) and body mass (kg) of all subjects were found using a precision stadiometer and a weighing scale (SECA 222 and 634, respectively, SECA Corp., Hamburg, Germany). All measurements were taken with the participants wearing only underwear.

The duration (ms) of the foot rockers was measured by employing the Freemed™ baropodometric platform (Freemed, SensorMedica, Rome, Italy), whose absolute agreement and consistency has been evaluated [19] using the following references: rocker 1 (R1) from the start of heel contact until forefoot contact; rocker 2 (R2) from forefoot contact until the heel rises; rocker 3 (R3) from the heel rising until the metatarsals rise from the ground; and rocker 4 (R4) from all the metatarsals being raised from the ground until the foot has completely left the ground (Figure 1).

The surface of the platform (i.e., 635 × 700 mm) offered an active sensor area of 500 × 600 mm. It recorded data at 350 Hz [19], and the calibration was carried out following the recommendations of the manufacturer. Eventually, it was linked to a laptop via USB. The manufacturer’s software (Freestep v. 2.00.013, SensorMedica, Rome, Italy) was utilized for data analysis. Only data obtained from the right foot were considered to avoid potential confounding factors (i.e., asymmetry) [20].

### 2.4. Tendon Morphology Characteristics

High-resolution ultrasound images were acquired in B-mode utilizing a linear probe with a frequency range of 5–16 MHz (LOGIQ S7 EXPERT, General Electric, Germany, 2013). Both longitudinal and transversal perspectives of the AT, PT, and PF were captured. According to a recent review, ultrasound measurements of tendon dimensions exhibit re-liability, encompassing both relative and absolute aspects [21].

To evaluate the AT, participants assumed a prone position with extended knees and feet outside the bed, maintaining the ankle in a neutral position. Tendon thickness and CSA were assessed using a reference point positioned 3 cm proximal to the tendon insertion in the calcaneus bone, as determined by the ultrasound device [22]. The images were taken with a depth of 2 cm and with the focus located at 0.5 cm [22].

The PT was measured with subjects in a supine position, with both knees bent at 30°; a reference of 1 cm distal to the lower pole of the patella was used to assess the tendon thickness and CSA [22]. The images were taken with a depth of 3 cm and with the focus located at 0.5 cm [22].

The ultrasound examination involved participants adopting a prone position with extended knees, neutral ankle position and toes extended against the bed surface. A reference point, identified by the ultrasound device, was located vertically from the anterior edge of the plantar surface of the calcaneus bone to the anterior edge of the plantar fascia to measure the thickness of the plantar fascia [22]. The images were taken with a depth of 3 cm and with the focus located at 1 cm [22].

A 12 MHz frequency and 100 dB gain were consistently applied for all structures. Each measurement was recorded twice by a proficient researcher with over ten years of experience. After data collection, thickness and CSA were analyzed using ImageJ software, version 1.53 t (NIH, Baltimore, MD, USA) [23], employing the “free hand” tool for CSA determination.

### 2.5. Statistical Analysis

Mean and standard deviation (SD) are used to present descriptive data. Kolmogorov–Smirnov’s and Levene’s tests were employed to confirm the data normality distribution and homogeneity of variances, respectively (*p* > 0.05). Three different k-means cluster analyses were proposed: (i) regarding CSA and thickness of the PT; (ii) considering CSA and thickness of the AT; and (iii) thickness of the PF. Thus, within each cluster for each tendon, participants were divided into two thickness groups (higher tendons group (HG) and lower tendons group (LG)). Data from HG and LG were compared for all the tendons by a pairwise mean comparison (*t*-test). The magnitude of the differences between values was interpreted by employing the Cohen’s d effect size (ES) (between-group differences) and reported as trivial (<0.19), small (0.2–0.49), medium (0.5–0.79), and large (≥0.8) [24]. The SPSS 28.0 (SPSS Inc., Chicago, IL, USA) was utilized to perform all statistical analyses, and statistical significance was accepted at an alpha level of 0.05.

## 3. Results

The between-group comparison of the foot rockers regarding PT (Table 2) showed a significant difference (*p* = 0.035) in the mean value of the duration of R1 between HG (190.77 ± 51.94 ms) and LG (156.82 ± 42.01 ms). The ES of the difference was reported as medium (0.718). When the duration of R1 relative to the total duration of the gait cycle was analyzed, a significant difference (*p* = 0.040) was found between the HG (31.13 ± 9.11) and the LG (25.54 ± 6.59). The ES of this difference was reported as medium (0.703).

Regarding AT (Table 3), no significant difference was found between HG and LG in the relationship between AT morphology and foot rockers in absolute duration or relative to the total duration of the gait. However, the ES of the difference in both absolute (0.610) and relative (0.569) duration for AT and R1 was medium.

Table 4 exhibits the values for the between-group comparison (HG vs. LG) of the foot rockers regarding PF. No significant difference was found between HG, LG and PF morphology and any of the rockers. The ES of the difference in both absolute and relative (0.569) duration was small for R1, R2 and R4 (0.231–0.472) and trivial for R3 (<0.086).

## 4. Discussion

This study aimed to answer the following research question: what is the relationship between the morphology of the patellar tendons (PT) and Achilles tendons (AT) and plantar fascia (PF) and the duration of the rockers? A significant correlation between the PT thickness and the R1 duration resulting in longer durations of R1 for participants with a thicker PT was found in the present study. On the contrary, the structures of the AT and PF seem to have no influence on the duration of the rockers.

The importance of muscle function in the development of functional rockers has been already examined [25]. During R1, both the tibialis anterior and the extensor digitorum longus muscles play essential roles as they eccentrically control the drop of the foot while flattening [25]. However, in the aforementioned study, the behavior of the quadricep muscles during this process was not considered. It is known that during the initial phase of the stance phase, where R1 is included, the degrees of knee flexion increase, requiring the eccentric activation of the muscle–tendon unit of the quadriceps [26].

Based on the results of the present study, it seems that for individuals with a thicker PT, these tendons would elongate longer while absorbing energy within the SSC, which could explain the longer duration of R1 reported here. These findings are supported by a previous study completed with patients with a diabetic foot [14]. Patients with a diabetic foot tend to have thicker tendons [15] and longer stance phases when compared with healthy individuals [27]. However, it is worth mentioning that in patients with a diabetic foot, apart from tendinous morphological differences, neurological aspects need to be considered when explaining possible gait alterations [25]. To the best of the authors’ knowledge, no previous study has evaluated the relationship between the structure of the PT, AT and PF and the duration of the foot rockers. Additionally, although non-significant, the results of this study showed a longer duration of R1 in the group of participants with thicker PT and AT, with a medium ES (>0.569).

The identified correlation between the thickness of the patellar tendon (PT) and the duration of the first rocker (R1) in walking among healthy adults holds substantial clinical implications. Understanding this relationship provides valuable insights into the biomechanical factors influencing gait dynamics. Particularly for individuals with a thicker PT, the elongation of these tendons during the stretch–shortening cycle (SSC) suggests a potential mechanism contributing to the observed prolonged R1.

The practical significance of this study lies in its contribution to refining our comprehension of the nuanced interplay between tendon morphology and walking gait. The observed association between PT thickness and R1 duration prompts considerations for rehabilitation and intervention strategies. Tailoring interventions aimed at optimizing tendon function, especially the PT, may offer avenues for modulating gait patterns, potentially benefiting individuals with gait abnormalities or those undergoing rehabilitation. Furthermore, the medium effect size observed in the group with thicker PT and AT warrants attention, suggesting that even in the absence of statistical significance, there may be practical relevance to explore further in subsequent research.

There are some limitations to consider when analyzing the findings reported here. First, only healthy males participated. This fact may constitute a bias when extrapolating the results to other populations (i.e., women, children, or pathological populations), but it helped to standardize the sample. No comparison with pathological (i.e., gait pathologies) or sport (i.e., runners) populations has been carried out here. Long-distance runners tend to show larger dimensions in terms of CSA and thickness in the PT, AT and PF given the amount of load supported by these tendons [28]. As it seems that a thicker PT significantly correlates with a longer R1, we might hypothesize that long-distance runners would exhibit longer R1 duration. However, this should be considered in future studies.

## 5. Conclusions

The findings of this study demonstrate a link between PT thickness and the duration of rocker 1 in healthy adults, elucidating a noteworthy distinction. Individuals with a thicker PT exhibited a substantially prolonged rocker 1 compared to those with thinner tendons. This pivotal revelation underscores the imperative role of tendon morphology in influencing the temporal aspects of gait, particularly the functional rockers. Consequently, the incorporation of tendon morphology assessments emerges as a crucial consideration in the comprehensive analysis of gait biomechanics, offering a perspective that enhances the precision and relevance of such evaluations.

## Figures and Tables

**Figure 1 healthcare-11-03142-f001:**
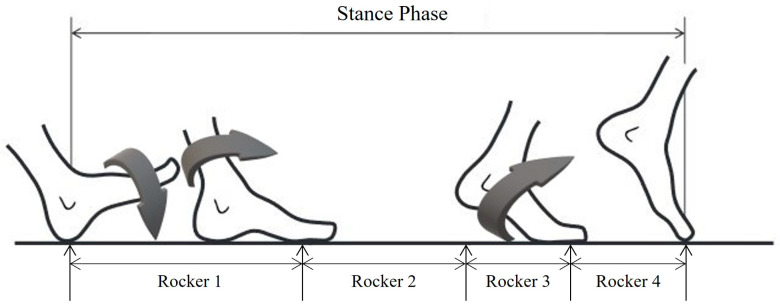
Functional foot rocker diagram for analysis. Adapted from Núñez-Trull et al., 2023 with permission.

**Table 1 healthcare-11-03142-t001:** Descriptive characteristic of the participants (mean, ±SD).

Variable	
Age (years)	28.97 (7.7)
Height (m)	176.40 (6.3)
Body mass (kg)	71.66 (7.8)
BMI (Kg·m^−2^)	23.02
PT-Thickness (mm) PT-CSA (mm^2^)	3.73 (0.7) 94.99 (22.8)
AT-Thickness (mm) AT-CSA (mm^2^)	5.19 (0.6) 53.68 (8.6)
PF-Thickness (mm)	2.66 (0.4)

BMI: Body Mass Index; CSA: cross-sectional area; PT: patellar tendon; AT: Achilles tendon; PF: plantar fascia.

**Table 2 healthcare-11-03142-t002:** Between-group comparison (HG vs. LG) of the duration the foot rockers regarding thickness and CSA of the patellar tendon.

	HG (n = 22)	LG (n = 17)	*p*-Value ^	ES (*d*)
R1 (ms)	190.77 ± 51.94	156.82 ± 42.01	0.035 *	0.718
%R1 (% gait cycle)	31.13 ± 9.11	25.54 ± 6.59	0.040 *	0.703
R2 (ms)	107.5 ± 51.75	110.59 ± 50.45	0.853	0.060
%R2 (% gait cycle)	17.44 ± 8.41	17.87 ± 8.12	0.875	0.052
R3 (ms)	300.82 ± 64.14	321.76 ± 50.04	0.274	0.364
%R3 (% gait cycle)	48.50 ± 8.68	52.56 ± 8.8	0.158	0.464
R4 (ms)	18.55 ± 16.63	24.76 ± 14.34	0.227	0.135
%R4 (% gait cycle)	2.93 ± 2.6	2.6 ± 2.3	0.179	0.134

R1: rocker 1; R2: rocker 2; R3: rocker 3; R4: rocker 4; ms: milliseconds; HG: group with thicker tendons; LG: group with thinner tendons; ES: Cohen’s d effect size. ^ *t*-test. * *p*< 0.05.

**Table 3 healthcare-11-03142-t003:** Between-group comparison (HG vs. LG) of the duration of the foot rockers regarding thickness and CSA of the Achilles tendon.

	HG (n = 20)	LG (n = 19)	*p*-Value ^	ES (*d*)
R1 (ms)	190.40 ± 52.46	160.79 ± 44.16	0.065	0.610
%R1 (% gait cycle)	30.98 ± 9	26.29 ± 7.39	0.084	0.569
R2 (ms)	105.55 ± 35.28	112.32 ± 63.67	0.682	0.131
%R2 (% gait cycle)	16.9 ± 5.65	18.3 ± 10.32	0.626	0.168
R3 (ms)	301.45 ± 58.62	318.89 ± 58.88	0.360	0.296
%R3 (% gait cycle)	48.63 ± 8.60	52 ± 9	0.239	0.382
R4 (ms)	21.65 ± 17.92	20.84 ± 13.65	0.876	0.050
%R4 (% gait cycle)	3.4 ± 2.7	3.4 ± 2.3	0.984	0.000

R1: rocker 1; R2: rocker 2; R3: rocker 3; R4: rocker 4; ms: milliseconds; HG: group with thicker tendons; LG: group with thinner tendons; ES: Cohen’s d effect size. ^ *t*-test

**Table 4 healthcare-11-03142-t004:** Between-group comparison (HG vs. LG) of the duration of the foot rockers regarding thickness of the plantar fascia.

	HG (n = 18)	LG (n = 21)	*p*-Value ^	ES (d)
R1 (ms)	169.72 ± 45.75	181.33 ± 54.33	0.479	0.231
%R1 (% gait cycle)	27.44 ± 8.1	29.77 ± 8.89	0.401	0.273
R2 (ms)	121.5 ± 49.45	98 ± 50.1	0.150	0.472
%R2 (% gait cycle)	19.54 ± 7.92	15.99 ± 8.22	0.180	0.439
R3 (ms)	312.72 ± 59.37	307.57 ± 59.36	0.789	0.086
%R3 (% gait cycle)	50.06 ± 7.86	50.46 ± 9.8	0.891	0.045
R4 (ms)	18.72 ± 14.16	23.43 ± 16.76	0.360	0.303
%R4 (% gait cycle)	2.96 ± 2.29	3.78 ± 2.67	0.312	0.329

R1: rocker 1; R2: rocker 2; R3: rocker 3; R4: rocker 4; ms: milliseconds; HG: group with thicker tendons; LG: group with thinner tendons; ES: Cohen’s d effect size. ^ *t*-test.

## Data Availability

The data presented is available upon express request.

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
