# Peer review of "Does Lower-Limb Tendon Structure Influence Walking Gait?"

_healthcare, 2023, doi:10.3390/healthcare11243142_

Round 1
Reviewer 1 Report
Comments and Suggestions for Authors
This is an observational study carried out in 39 healthy men that assesses the relationship between the morphology of the patellar and Achilles tendons, and the plantar fascia with respect to the functional rockers. Although it is very simple and methodologically poor, it is an interesting approach.
I would like to point out some aspects to the authors:
Can you provide the ethics committee approval reference number?
The sample selection method and age range are not clear. Why were women not included in the study in order to study whether there are differences regarding gender?
In line 101 it is stated that "A sample large enough to represent the target population of this study was selected." How was this selection made?
The bibliography must be reviewed. For example, if you use this method to indicate the pages in reference 25: 640-7, why do you indicate the pages in reference 26 as 331-351?
Author Response
Reviewer #1: This is an observational study carried out in 39 healthy men that assesses the relationship between the morphology of the patellar and Achilles tendons, and the plantar fascia with respect to the functional rockers. Although it is very simple and methodologically poor, it is an interesting approach.
Thank you for taking the time to review our manuscript. We have made all the suggested changes, which are highlighted in yellow. We believe that your input has greatly enhanced the overall quality of our work, and we hope that you are satisfied with the revised manuscript.
I would like to point out some aspects to the authors:
Can you provide the ethics committee approval reference number?
Ethics committee approval reference number has been added to the text. Thanks.
The sample selection method and age range are not clear. Why were women not included in the study in order to study whether there are differences regarding gender?
We understand that it could have been more interesting to have a sample composed of men and women, but the number of women evaluated was very small and, seeking to homogenize the sample as much as possible, it was decided to use only men to show consistent results in, at least, men. Thanks for your comment, we will consider this fact for future studies.
In line 101 it is stated that "A sample large enough to represent the target population of this study was selected." How was this selection made?
Thanks for this comment. The next explanation has been added to the text. “Sample size power calculation was executed using G*POWER 3.1.9.7 (University of Dusseldorf, Dusseldorf, Germany). The following structure was used based on the analysis: Means differences between two independent means (matched pairs); A priori. Effect size dz= 0.5; α error prob =0.05; power (1-β error prob) =0.90. The result showed a suitable total sample size of 39 athletes for actual high power (90.25%)”
The bibliography must be reviewed. For example, if you use this method to indicate the pages in reference 25: 640-7, why do you indicate the pages in reference 26 as 331-351?
Bibliography has been reviewed and corrected. Thanks.
Reviewer 2 Report
Comments and Suggestions for Authors
The study aimed to analyze the relationship between the morphology of the patellar and Achilles tendons, and plantar fascia with respect to the duration the rockers. Although it is a very interesting study, there are several significant concerns that need to be addressed.
1. There are several areas in the Abstract section that require improvement. 1) please improve the background information to better emphasize the novelty and significance of the research. 2) there is less description about the research methods here, for example, some basic information of the participants should be added here. In addition, please explain how the relationship between the morphology of the soft tissue and the duration of the rockers was determined. 3) The conclusion should be further strengthened based on the main findings of the study; 6) it would be beneficial to improve the quality of the keywords used to make it easier for researchers and other interested parties to locate information relevant to the research topic.
2. In the Introduction section, the authors described their objective to analyze the relationship between the morphology of the patellar and Achilles tendons, and plantar fascia with respect to the duration the rockers. However, the explanation of the research gap was insufficient, and the novelty of the study was not clearly communicated. Merely stating that no research has been conducted on this topic does not necessarily indicate the significance of the study. It is essential to explain what new insights this study can bring to the existing knowledge in this field and how it is relevant to clinical and practical situations. For instance, previous studies have extensively investigated the impact of these soft tissues on motion features and the potential risk of foot injury (https://doi.org/10.1080/10255842.2023.2268231; https://doi.org/10.1177/09544119221085795). It is recommended that the authors include this information at the outset of their study to further emphasize its clinical and practical significance by citing the abovementioned recent research.
3. The Methods section of the study provides limited information on the participants and data analysis. To improve the clarity and comprehensiveness of this section, the following suggestions are recommended. 1) The authors did not clarify whether the sample size of this trial was calculated or just set according to experience; 2) the inclusion criteria were relatively broad, which kind of people did this study aim at? 3) “This procedure was repeated for 3 min.”, for how many trials each participant? 4) it is suggested to use a figure to clearly explain the definition of Rocker 1 to Rocker 4. 5) “Only data obtained from the right foot were considered to avoid potential confounding factors”, dominant foot?
4. For the results of this study, 1) it is suggested that a table could be used to clearly present the basic information of the participants and the morphology of these soft tissues; 2) it is suggested that this session could be further divided into several sub-session based on different part of the soft tissue.
5. Discussion, it is suggested that more comparisons should be made with previous related research based on the findings of this study. Meanwhile, the clinical and practical significance of this study should be further highlighted in this session.
6. The conclusion should be further strengthened based on the main findings of the study.
Author Response
Reviewer #2: The study aimed to analyze the relationship between the morphology of the patellar and Achilles tendons, and plantar fascia with respect to the duration the rockers. Although it is a very interesting study, there are several significant concerns that need to be addressed.
Thank you for taking the time to review our manuscript. We have made all the suggested changes, which are highlighted in yellow. We believe that your input has greatly enhanced the overall quality of our work, and we hope that you are satisfied with the revised manuscript.
- There are several areas in the Abstract section that require improvement. 1) please improve the background information to better emphasize the novelty and significance of the research. 2) there is less description about the research methods here, for example, some basic information of the participants should be added here. In addition, please explain how the relationship between the morphology of the soft tissue and the duration of the rockers was determined. 3) The conclusion should be further strengthened based on the main findings of the study; 6) it would be beneficial to improve the quality of the keywords used to make it easier for researchers and other interested parties to locate information relevant to the research topic.
The Abstract section has been rewritten and improved regarding the reviewer considerations. Thanks.
- In the Introduction section, the authors described their objective to analyze the relationship between the morphology of the patellar and Achilles tendons, and plantar fascia with respect to the duration the rockers. However, the explanation of the research gap was insufficient, and the novelty of the study was not clearly communicated. Merely stating that no research has been conducted on this topic does not necessarily indicate the significance of the study. It is essential to explain what new insights this study can bring to the existing knowledge in this field and how it is relevant to clinical and practical situations. For instance, previous studies have extensively investigated the impact of these soft tissues on motion features and the potential risk of foot injury (https://doi.org/10.1080/10255842.2023.2268231; https://doi.org/10.1177/09544119221085795). It is recommended that the authors include this information at the outset of their study to further emphasize its clinical and practical significance by citing the abovementioned recent research.
Thank you very much for your suggestion. The authors consider that both the explanation of the research gap and the novelty of the study is clearly stated (lines 72-75, 76-86). On the other hand, the authors thank the reviewer for the two interesting references provided, which have been added to the introduction (highlighted in yellow).
- The Methods section of the study provides limited information on the participants and data analysis. To improve the clarity and comprehensiveness of this section, the following suggestions are recommended. 1) The authors did not clarify whether the sample size of this trial was calculated or just set according to experience; 2) the inclusion criteria were relatively broad, which kind of people did this study aim at? 3) “This procedure was repeated for 3 min.”, for how many trials each participant? 4) it is suggested to use a figure to clearly explain the definition of Rocker 1 to Rocker 4. 5) “Only data obtained from the right foot were considered to avoid potential confounding factors”, dominant foot?
Thanks for this comment. 1) The next explanation has been added to the text. “Sample size power calculation was executed using G*POWER 3.1.9.7 (University of Dusseldorf, Dusseldorf, Germany). The following structure was used based on the analysis: Means differences between two independent means (matched pairs); A priori. Effect size dz= 0.5; α error prob =0.05; power (1-β error prob) =0.90. The result showed a suitable total sample size of 38 athletes for actual high power (90.25%)”. 2) The objective of the study was to evaluate healthy adults. For this reason, those subjects who had not suffered an injury to the lower limb during the last 6 months and who did not observe a pathological gait during the evaluation were included in the study. This information is included in the methods section. 3) This procedure was repeated for 3 min, so that the subject walked on the examination walkway as many times as possible at a comfortable speed. This information has been included in the text. 4) New Figure (Figure 1) has been included. 5) To homogenize data collection and avoid potential confounding factors only data obtained from the right foot were considered, without considering whether it was the dominant foot.
- For the results of this study, 1) it is suggested that a table could be used to clearly present the basic information of the participants and the morphology of these soft tissues; 2) it is suggested that this session could be further divided into several sub-session based on different part of the soft tissue.
Thanks for this comment. 1) New table (Table 1) has been included regarding the reviewer suggestion. 2) The comparison in the HG and LG groups was carried out by clustering in relation to the CSA and thickness variables, for this reason it is shown in the text in those divisions.
- Discussion, it is suggested that more comparisons should be made with previous related research based on the findings of this study. Meanwhile, the clinical and practical significance of this study should be further highlighted in this session.
Thanks for this comment No more similar studies have been found with which to compare the results of this study. However, following the author's instructions, the discussion section has been improved by expanding the clinical and practical implications.
- The conclusion should be further strengthened based on the main findings of the study.
The conclusion has been rewritten and strengthened. Thanks.